# Genome-Wide Association Study in Bread Wheat Identifies Genomic Regions Associated with Grain Yield and Quality under Contrasting Water Availability

**DOI:** 10.3390/ijms231810575

**Published:** 2022-09-12

**Authors:** Nikolai Govta, Iris Polda, Hanan Sela, Yafit Cohen, Diane M. Beckles, Abraham B. Korol, Tzion Fahima, Yehoshua Saranga, Tamar Krugman

**Affiliations:** 1Institute of Evolution, Department of Evolutionary and Environmental Biology, University of Haifa, Haifa 3498838, Israel; 2Smith Institute of Plant Science & Genetics in Agriculture, The Hebrew University of Jerusalem, Rehovot 7632706, Israel; 3Institute of Evolution, University of Haifa, Haifa 3498838, Israel; 4Agricultural Research Organization, Volcani Center, Institute of Agricultural Engineering, Beit Dagan 7505101, Israel; 5Department of Plant Sciences, University of California, Davis, CA 95616, USA

**Keywords:** *Triticum aestivum* L., water-limited, GWAS, grain protein content (GPC), grain yield, marker trait association (MTA), quantitative trait loci (QTL)

## Abstract

The objectives of this study were to identify genetic loci in the bread wheat genome that would influence yield stability and quality under water stress, and to identify accessions that can be recommended for cultivation in dry and hot regions. We performed a genome-wide association study (GWAS) using a panel of 232 wheat accessions spanning diverse ecogeographic regions. Plants were evaluated in the Israeli Northern Negev, under two environments: water-limited (D; 250 mm) and well-watered (W; 450 mm) conditions; they were genotyped with ~71,500 SNPs derived from exome capture sequencing. Of the 14 phenotypic traits evaluated, 12 had significantly lower values under D compared to W conditions, while the values for two traits were higher under D. High heritability (*H*^2^ = 0.5–0.9) was observed for grain yield, spike weight, number of grains per spike, peduncle length, and plant height. Days to heading and grain yield could be partitioned based on accession origins. GWAS identified 154 marker-trait associations (MTAs) for yield and quality-related traits, 82 under D and 72 under W, and identified potential candidate genes. We identified 24 accessions showing high and/or stable yields under D conditions that can be recommended for cultivation in regions under the threat of global climate change.

## 1. Introduction

Wheat is a major source of starch and energy and essential compounds, such as proteins, minerals, vitamins, and dietary fibers, which are beneficial for human health [1]. The demand for wheat, both for food and animal feed, is expected to increase due to world population growth, estimated to be ~9.8 billion people by 2050 (World Population Prospects: The 2017 Revision | UN DESA Publications (https://desapublications.un.org/publications) (accessed on 1 June 2017). Hence, to meet the challenge of feeding the growing world population, wheat production must grow by almost a billion tons by 2050 (from the current production of 2.1 billion tons). Instead, the growth rate of wheat yields has stagnated since the 1990s, mostly due to the severity of negative environmental factors such as drought or high temperatures, which are expected to worsen with global climate change [2,3,4,5].

Drought affects more than 42% of the worldwide wheat production area, and even small deviations in water availability can significantly decrease grain yield [6,7,8,9,10]. Furthermore, high temperatures combined with drought during grain formation and grain filling periods can reduce productivity and grain quality [11,12,13]. For example, heat and drought in Africa and the Mediterranean can cause a decrease in grain yields by 24% [14], making it challenging to ensure food security in hot and arid regions such as parts of sub-Saharan Africa and southeastern and western Asia (FAO, 2017; 2020). The occurrence of spatial environmental heterogeneity requires a refined understanding of plant responses to adverse variable conditions to enable the breeding of resilient wheat varieties that can adapt to future climatic conditions [15,16,17]. High grain yield (GY) and grain protein content (GPC) are key economic components that underlie successful wheat cultivars. GY and GPC are complex quantitative traits, controlled by multiple gene loci that are influenced by the environment. GY is determined by developmental processes including tillering, floral development, plant architecture, phenology, plant height, and assimilate partitioning. Total GY integrates several yield components, including the number of grains per spike, spikelet per unit area, number of productive shoots, and thousand kernel weight [16,18,19,20]. GPC is regulated by a complex genetic system and is affected by environmental factors such as drought and heat, and by agricultural practices such as mineral fertilizer application since it is highly dependent on nitrogen availability. GPC commonly shows a negative correlation with GY [21]. High transpiration limits the rate of nitrogen mineralization and reduces its uptake [22]. This in turn inhibits the deposition of starch, which leads to shriveled grain and changes in the balance between glutenin and gliadin proteins, with repercussions for quality [5]. 

Quantitative trait loci (QTL) mapping can be used to dissect the genetic loci underlying complex traits such as GY and GPC in wheat. For many years, QTL mapping was applied to segregating populations derived from bi-parental crosses, and associations between their loci and phenotypes of interest were investigated. The resulting QTL maps provide the genetic locations of QTL regions, allelic effect, epistatic interactions, and information on molecular markers flanking promising QTLs that can be used for breeding by marker-assisted selection (MAS) [23,24]. QTL mapping based on bi-parental populations may suffer from limited recombination events and/or low polymorphisms between the parents. However, using a wild progenitor as a parent in a cross can increase the number of polymorphic markers and improve QTL mapping [25,26]. During the last two decades, genome-wide association studies (GWAS) have become a valuable tool for dissecting the genetic architecture of complex traits in different crops [27,28,29,30]. Genomic technologies and the availability of whole-genome sequencing of tetra- and hexaploid wheat provide information on thousands of SNP markers that can be used for high throughput genotyping, identification of marker-trait associations (MTAs), and identification of putative candidate genes that underlie the traits [27,31,32]. GWAS was used to dissect quality, biotic, and abiotic stresses, and agronomic and yield-related traits in wheat, rice, barley, and maize [33,34,35,36,37]. These studies demonstrated that GWAS performed with diverse germplasm could improve QTL mapping resolution. GWAS was also used to identify 15 wheat rust resistance QTLs, and 36 novel mineral concentrations QTLs for controlled introgression in breeding programs [38]. Furthermore, advanced statistical-genetics models that consider population structure and family relations are embedded in GWAS packages [39,40]. Using various GWAS models, e.g., MLM [41,42,43], MLMM [44,45,46], and FarmCPU [47] enabled a better understanding of the genetic determinants of cassava starch paste properties and helped with clonal selection for starch quality. Candidate genes that underlie valuable traits (i.e., either revealed by QTL mapping or by GWAS) can be identified and cloned based on their physical location on available whole genome sequencing [23,31,48], or manipulated by gene editing (CRISPR/Cas9 system) to enhance plant breeding efforts [49,50]. For example, editing *TaGASR7*, *TaGW2*, and *TaLOX2* in hexaploid wheat, and *TdGASR7* in tetraploid durum wheat, was used to detect relationships between these loci and grain yield components [23]. 

In this study, we used a wheat panel of 232 accessions from diverse ecogeographic regions (Europe, Asia, Africa, the Mediterranean, Oceania, and America) [51]. The panel was evaluated in the field under contrasting water availability and genotyped with 71,571 SNPs derived from exome capture sequencing. The objectives of our study were to genetically dissect the response to water stress in bread wheat by GWAS, and, to identify genetic loci in the bread wheat genome that would influence yield stability and quality under water stress. Based on the phenotypic response of the wheat panel to water stress, to identify accessions can be recommended for cultivation in dry and hot regions which are under threat of climate change.

## 2. Results

### 2.1. Descriptive Statistics of Phenotypic Traits

The data of 17 phenotypic traits were recorded in the 232 wheat accessions grown under two water regimes, i.e., the D and W conditions (Table 1). 

Comparative characteristics of the distributions of phenotypic traits and correlation coefficients under the two contrasting water conditions are presented in Appendix A. Analysis of variance (ANOVA) revealed significant (*p* < 0.05) differences between D and W for all the analyzed traits (Appendix A). Three of them (OP, LL, and LW) were excluded that were not significantly different between D and W were excluded from further analysis. Of the 14 tested traits, the values of 12 were lower under D (*p* < 0.05) as compared with W. The greatest reductions of trait values by D were found in GY (27%), GWpS (22%), GPC (17%), PedL (15.8%), LA (13.8%), TGW (11.7%), and PH (10%). The values of two traits—SL and SLW—were higher by 7% and 8% in D as compared with W, respectively (Figure 1).

Significant positive correlations were found between values of GY and yield components (i.e., SW, GWpS, GNpS, NSpP, TGW, GFE) under both D and W conditions (0.32 to 0.74, *p* = 0.001 in D; and 0.38 to 0.75, *p* = 0.001 in W). Furthermore, significant negative correlations were found under both water regimes between GY and PH, DH, and GPC (−0.32 to −0.57, *p* = 0.001 in D; and −0.43 to −0.60, *p* = 0.001 in W). A positive correlation was found between GY and SLW under W, while a negative correlation was found with PedL (Appendix A). 

Estimates of heritability (*H*^2^) for the 14 traits measured in both conditions, ranged from 0.10 to 0.99 under W and from 0.12 to 0.75 under D. Under the W conditions, high *H*^2^ (>0.70) was found for six traits (GY; SW; GWpS; PH; PedL, and DH); moderate *H*^2^ (0.20–0.50) was observed for the following traits (GNpS; NSpP; TGW; GFE; LA; SLW, and SL). Low *H*^2^ (<0.20) was found for GPC in both water regimes. Under the D condition, *H*^2^ of more than 0.50 was found for six traits (GY; SW; GWpS; NSpP; PH; and PedL). Moderate levels of *H*^2^ (0.20–0.50) were observed for the traits (GPC; GNpS; GFE; LA; SLW; SL and DH) and the lowest *H*^2^ (0.12) was obtained for TGW.

### 2.2. Canopy Temperature and Stress Index

A thermal image of wheat experimental plots was acquired at midday on April 20th, 2017. The temperature map (Figure 2a) demonstrated the temperature difference between the W and D conditions, as well as the wide variation within each treatment that shows non-stressed (with a surface temp scale of 23.5–31 °C) accessions also under D. In accordance with this, CWSI of the plots under the W treatment varied between 0.65–1.15 with an average of 0.8. In comparison, CWSI of the plots under the D treatment varied between 0.85–1.2 with an average of 1. Thus, both canopy temperatures and CWSI confirm the high level of water stress imposed on the D treatment, and the within-treatment diversity enabled the identification of accessions that were less affected by the stress (Figure 2c).

### 2.3. Association of Phenotypic Traits with Geographic Origins

Our wheat panel includes spring-type landraces, and modern cultivars of bread wheat that were bred and adapted to specific ecogeographic regions. With the recent threat of climate change (e.g., low precipitation and/or high temperature), there is a need to identify new cultivars that show higher plasticity under these conditions. It is also important to identify the most promising ecogeographic origin for drought-tolerant wheat germplasm. Although there is a large variation in ecology between and within continents, we further tested if the abilities of accessions to cope with water stress are related to their origin by continent. 

We found large variation in all phenotypic traits among accessions of the same ecogeographic origins (Europe, Asia, Mediterranean, Africa, America, and Oceania). However, we also found large differences between continents. Due to the large variation within each continent, not all differences between continents were statistically significant (Appendix A). Traits that were significantly different (*p* < 0.05) between origins under W and/or D, including (a) DH—were different between ecogeographic origins under both water regimes. African accessions showed the minimum DH under W and D. The average DH of the African and Mediterranean accessions under D was 16–19 days shorter than those from Europe and Asia. The longest DH of 115 days was found in the European accessions under W, and 113 days under D; (b) GY—the distribution of GY under both water regimes was different between African and European accessions and between Asian and the Mediterranean and with African accession. The highest average GY was obtained by the Mediterranean accessions. A minimal difference in GY between the two water regimes, showing high yield stability between environments, was found in the African accessions (599.18 g/m^−2^. vs. 681.17 g/m^−2^), while maximal reduction of GY between D and W was found in the American accessions (532.6 g/m^−2^ vs. 745.6 g/m^−2^). GY in African accessions was on average, 31% (*p* < 0.001) higher than Asian accessions in D; (c) PH- significant differences were found in PH measured at the two water regimes in the European accessions (96.62 vs. 109.63) and Asian accessions (109.38 vs. 122.17). PH of European accessions under D was lower by 12% (*p* < 0.05) than Asian accessions. (d) SW of the Mediterranean accessions in D was 24% higher (*p* < 0.05) than that of the Asian accessions. Under the W condition, the maximum value of SW was found in the Mediterranean, and the minimum in the Asian accessions (3.22 vs. 2.55, *p* < 0.05); The small group of eight American accessions did not differ significantly in SW between conditions; (e) TGW of the Mediterranean accessions under W was 12% higher (*p* < 0.05) than that of the European accessions; (f) GPC—the maximum discrepancy between treatments in GPC was observed in the American accessions (12.46 vs. 15.15, *p* < 0.001) while it was minimal for the Asian accessions (10.92 vs. 13.06, *p* < 0.05). Significant changes between the two water regimes were found also in European accessions (11.28 vs. 13.72, *p* < 0.001). The GPC of the Mediterranean accessions was reduced under D by about 16% (12.6 vs. 15.0, *p* < 0.05).

### 2.4. Identification of Drought-Tolerant and Stable Accessions

We used PCA analysis to select accessions that have high GY under each water regime and accessions that show high yield stability across environments. The first two principal components together accounted for 49.08% and 47.52% of the variance under W and D conditions, respectively (Figure 3).

Under W conditions, PC1 explained 31.89% of the variation in positive (GY, SW, GWpS, GNpS) and negative (PH, PedL, and DH) traits, respectively. PC2 explained 17.19% of the variation and was positively loaded by DH and NSpP, and negatively loaded by (PH, PedL, and TGW). Under the D conditions, PC1 explained 29.58% of the variation in the positively and negatively correlated traits. PC2 explained 17.94% of the variation and was positive for (PH and PedL) and negative for (DH and GNpS). The relationship in the form of interactions between traits was confirmed by the results of the correlation network (Appendix A), which show for example that under D, a long period of DH will negatively affect GY, SW, and GNpS.

By calculating the cosine squared for GY (FC > 0.70) correlations, we identified 15 accessions that performed the best under D and can be regarded as drought tolerant. In addition, nine accessions showed excellent high and stable yields in both water regimes, five of them were from Israel. Altogether, these 24 drought-tolerant and stable accessions mostly originated from relative narrow latitudes ranging from 28° to 38°, 20 of them being from dry environments, in the Mediterranean, Africa, and Asia, and one accession originating from Mexico (latitudes ranging from 30° to 16°) present at (Appendix A and Figure 4).

Under D, the maximum average of GY of (771.49 g/m^2^ ± 59.88 g/m^2^) was found among 12 accessions from the Mediterranean, six from Israel, three from Turkey, and two from Syria, with an average PH of (88 ± 8.56 cm) and DH (94 ± 3.52). The average GY of the African and Asian drought-resistant accession ranged from 599.18 g/m^2^ to 765.35 g/m^2^. We found additional 16 accessions that had high yield under W, six of them originated from the Mediterranean while the rest, were from diverse ecogeographic regions from wider latitude distributions, including Switzerland, Germany, and Bulgaria. In total, we identified 40 accessions that were grown in the hot and dry environment at the Israel Northern Negev which can be recommended for cultivation in regions under threat of adverse climatic conditions including high temperatures and low precipitation. 

### 2.5. Description of GWAS. Genotyping and Genetic Structure of the Wheat Panel

We used a total of 71,571 SNPs for genotyping and further GWAS analysis, 42.2% of the SNPs were mapped to the A genome, 49.8% to the B genome, and only 7.9% were mapped to the D genome (Figure 5a).

SNP markers were not uniformly distributed among wheat chromosomes. As a result, wheat chromosomes differ in length and the number of mapped SNPs. The lowest coverage was found on all chromosomes of the D genome, of which Chr. 6D had the lowest length size (486.2 Mb) and the greatest on Chr. 3B (828 Mb) (Figure 5b); the minimum number of SNPs (5338) were mapped to chromosomes of group 4, of which Chr. 4B had the lowest number, i.e., 1943, while the maximum number of SNPs were mapped to the Chr. 2 group (13,248).

### 2.6. Population Structure and Kinship Analysis

The results of STRUCTURE analysis based on markers with minor allelic frequency (MAF) > 0.05 for the wheat accessions showed that ΔK was highest at K = 4, indicating the presence of four main clusters in the population (Figure 6a).

Although some impurities were present within clusters, a very clear separation was obtained. Thus, the red area in Figure 6a included 51 accessions consisting of 90.2% Asian, with the remaining 5 accessions unevenly distributed over clusters two, three, and four. The green area of 37 accessions was composed of 35% African, 54.5% Asian, and 32% European accessions, respectively. The blue area of 116 accessions was composed of 6% African, 14.6% American, 23.27% Asian, 49.13 European, and 5.17% Oceanic accessions. The yellow area consisted of 28 accessions, almost all (96.4%) were European accessions, with one Asian accession. The kinship matrix used in GWAS gave similar results to the STRUCTURE analysis (Figure 6b). For GWAS analysis, we used three PCs from the PCA as a fixed effect covariate in GAPIT to adjust for population structure. The first three PCs of marker data, which explained 66.71% of the total variance, were used to draw a 3D plot of the population structure (Appendix A).

### 2.7. Identification of Marker-Trait Associations (MTAs)

To identify the genetic loci associated with the agronomic traits under the D and W conditions, we performed GWAS for all phenotypic traits with 71,571 SNPs. Five models in GAPIT (GLM, MLM, MLMM, FarmCPU, and BLINK) were tested to determine the optimal model for each trait [23]. The quantile-quantile (QQ) plots were used to determine false associations. In the example presented in Figure 7a we show the five tested models for GY, of them we selected FarmCPU in which its *p*-values were closest to the diagonal line of expected *p*-values. 

The best model was selected based on no inflation of most of the *p* values in the QQ plot, and only a few points deviating upwards [23]. After analyzing all traits using the five models, FarmCPU was selected for most traits (i.e., GY, SW, GNpS, GWpS, NSpP, PH, PedL, DH), while the GLM model was used for GPC. The significant MTAs were selected based on FDR corrected *p*-values with a threshold of 0.05. A total of 154 highly significant MTAs dispersed on all chromosomes were found for nine traits. Of them, 82 MTAs were found under D, and 72 MTA were found under W conditions (Figure 8a). The complete list of MTAs identified for all traits under the two water regimes is presented in Appendix A.

Manhattan Plots were created for all measurements of traits and SNPs in the 232 accessions tested under the two water regimes (Figure 9).

Statistically significant MTAs are marked on the plots with red and green dots. Red dots describe MTAs identified at (FDR < 0.01), the green dots (FDR < 0.05). For five traits, (TGW, GFE, LA, SLW, and SL), we did not find significant MTAs at the FDR < 0.1 thresholds, and a less decrease of FDR did not affect the detection of MTAs. Under W conditions, no significant MTAs were found for GPC at the detection threshold of (FDR > 0.1), while 13 MTAs were found under D with an FDR of (0.08–0.1).

### 2.8. Identification of MTAs Clusters in QTL Regions

We identified regions that included clusters of adjacent MTAs for a few traits. These were regarded as QTL regions which may include genes with epistatic interactions (Appendix A). For example, Chr. 1A included 17 MTAs for GY, PedL, SW, NSpP, PH, DH, GNpS, and GPC, clustered in 11 QTL regions. QTL 1A.10 included MTAs for PH and GPC under D, and QTL 1A.11 included an MTA for GY under W conditions together with MTA for GPC under D. The 23 MTAs for GY (13 found under W and 10 under D) were located almost on all chromosomes except for 3A and 4A. Only one MTA on Chr. 5A (QTL cluster 5.11) is found near MTA for DH under W. On the contrary, of the 20 MTAs found for DH, nine were included in QTL regions or adjacent regions with MTA for one of yield components. For example, QTL 1A.6 includes MTAs for GNpS in the two water regimes, together with MTA for DH in W conditions; QTL 1A.8 included an MTA for DH, and in the adjacent QTL 1A.9, an MTA for SW, both QTLs were detected in the two water regimes. QTL region 5B.7 included an MTA for DH, the adjacent QTL 5B.8 included MTA for GNpP (under D). 

We found an asymmetric distribution of MTAs between the A- and B-genomes under D, with a much higher number of MTAs, found in the B genome under D (27 A vs. 48 B), respectively, as compared with those found under W conditions (35 A vs. 27 B) (Figure 8b). Furthermore, MTAs were not evenly distributed among the wheat chromosomes, five chromosomes carried more than 10 MTAs (1A had 17; 2B-11; 5B-16; 6B-20, and 7A-12) while three chromosomes carried only 5 MTAs. On Chr. 1B, 3A, 7B, and on Chr. 4A we found two MTAs. Chr. 6B was found as an important source of MTAs, containing 20 in total, 13 of which were detected under D. Furthermore, four adjacent QTLs in 6B (6B.10–6B.13) were enriched in MTAs for yield (GY and SW) under both conditions, and for NSpP and DH only under D. We did not find a correlation between the number of SNPs and MTAs found on each chromosome. Nevertheless, since genome D carried both the lowest number of SNPs—probably due to the low diversity of genome D and low number of MTAs—we believe that the number of MTAs found on genome D is underestimated (only seven under the two conditions) (Figure 8b).

### 2.9. Candidate Genes Identified in QTL Regions

We found 513 and 604 high confidence genes around the 154 MTAs associated with the traits (in a frame of ±250 Kb from the MTA), using their physical location on *Triticum aestivum* cv. Chinese Spring genome as a reference (Appendix A). Based on gene annotation (GO) and a literature search, we assembled a shorter list of 41 candidate genes potentially associated with abiotic stresses i.e., drought, dehydration, cold, or heat (Appendix A). Below are examples of candidate genes deduced from MTAs for GY and GPC. A total of 23 significant MTAs were found for GY on almost all chromosomes; nine were identified under D and 14 under W conditions, while 5 significant MTAs were found for GPC only under the D conditions.

#### 2.9.1. MTAs for GY under D

The most significant MTAs for GY were found in QTL regions 1A.11, 2A.2, 2D.1, 3B.3, 4B.5, 5A.5, 5B.11, 6B.4, 6B.11, and 7D.5 (Appendix A, Figure 9). SNP WTa_076e74 found on Chr. 6B.4 resides in a gene encoding a plant-type leucine-rich repeat-containing N-terminal protein. Leucine-rich repeat (LRrR) receptor-like kinase (RLK) proteins play key roles in various biological pathways. Hypersensitivity response was observed in the rice LRR-RLK1 (*OsGIRL1*) Oryza sativa gene, in response to salt and heat stress, while a hyposensitivity response was observed to osmotic stress [52,53]. A second SNP, WTa_07bcda on Chr. 6B.11 resides in a gene encoding a transcription factor—Zinc finger NHR/GATA-type. Zinc finger transcription factors negatively regulate stomatal closure by directly modulating genes associated with H_2_O_2_ homeostasis and define a novel DownSTream (DST)-mediated H_2_O_2_ signaling pathway. Loss of DST function increases stomatal closure and reduces stomatal density, resulting in increased drought and salt tolerance in rice [26]. SNP WTa_030a25 on Chr. 2D.1 resides in a gene encoding protein with the CRAL-TRIO lipid-binding domain. This domain is found in proteins containing a *SEC14* domain which was found to be associated with drought in wild barley [27], and a target of *miR1436* under heat stress in rice [28], and in a *ZmSEC14p* from maize (*Zea maize* L.) that was upregulated in response to cold, salt, and Abscisic acid (ABA) [29]. The deduced protein of the candidate gene for drought tolerance found in wild barley (*Hsdr4*) shows similarity to the rice Rho-GTPase-activating protein-like with a *Sec14* p-like lipid-binding domain [27]. Downstream of WTa_030a25 we found a gene (TraesCS2D02G506500) encoding transcription factor, with the AP2/ERF domain. The AP2/ERF genes constitute a large multigene family, involved in increased tolerance to salt, drought, and diseases [31]. SNP WTa_03c4ed in 3B.3 was close to the gene (TraesCS3B02G040700), encoding a cation efflux protein. Cation efflux protein genes play a critical role in many aspects of plant growth, development, signaling, and stress response and are responsible for maintaining PH homeostasis and ion concentration in all living organisms [25,33]. The SNP WTa_0525cf in 4B.5 resides in a gene annotated as Alpha-L-arabinofuranosidase, known to be involved in cell-type-specific cell wall structure [32]. Close to this SNP we found the gene (TraesCS4B02G132500) annotated as an NAC domain transcription factor, that is involved in response to water stress and senescing leaves (https://github.com/Borrill-Lab/WheatFlagLeafSenescence/blob/master/data/TFs_v1.1.csv (accessed on 29 October 2018).

#### 2.9.2. MTAs for GY under W

SNP WTa_008782 found on Chr. 1A.7 resides in a gene encoding a protein from a superfamily with RmlC-like cupin domains which are important storage proteins associated with plant development [48]. RmlC-like cupin superfamily proteins and the cupin family were increased under flood stress [33]. Cupin proteins are widely associated with roles in extracellular matrix modification and interactions with plant pathogens and with salt stress [34,35]. SNP WTa_01a98f on Chr. 2A.1 resides in a gene (TraesCS2A02G299400), encoding protein kinase domain wheat participates in the regulatory networks governing stress processes including cytoplasmic calcium oscillation for drought. They are important components of MAPK cascades that play a critical role in plant growth and development [29]. SNP WTa_067bd2 on Chr. 5B.11 resides in (TraesCS5B02G521100), encoding a protein with IQ- motif, EF-hand binding site known to have a key role in plant growth and development, as well as in stress signaling as an important second messenger calcium Ca^2+^ [36]. Stimuli such as plant hormones, gravity, light, cold, heat, drought, anoxia, salt, touch, injury, and attack by pathogens can quickly cause an increase in cytosol-free Ca^2+^ ([Ca^2+^]_cyt_) [37]. The 89 DEGs were found that are potential N-sensitive candidates and indicate broader associations between nitrate and calcium signaling [38]. SNPs WTa_0bda7d in 7D.5 resides in (TraesCS7D02G402600), annotated as cyclophilin-type peptidyl-prolyl cis-trans isomerase domain. Transcriptional changes in the gene encoding proteins in the cyclophilin-type peptidyl-prolyl cis-trans isomerase/CLD were an effective foundation for enhancing photosynthesis and CO_2_ tolerance in algae [54].

#### 2.9.3. MTAs for GPC under D

MTAs were located on Chr. 1B, and 2A, two on Chr. 2B, and two on Chr. 6B. The QTL 6B.7 resides at the same location as the high grain protein gene Gpc-B1 that was previously cloned [55,56]. SNP WTa_078f94 resides at locus TraesCS6B02G262000, encoding a protein with ATP-dependent peptidase activity. ATP-dependent peptidase activity increases the synthesis and accumulation of water-soluble carbohydrates (sucrose) through glycolysis, providing an important mechanism for water conservation in plants during drought stress [52]. The SNP WTa_02013f on Chr. 2B resides in a gene encoding a zinc ion binding protein. Zinc finger ion binding proteins play a key role in resistance to abiotic stress; the *ZAT18* gene positively regulated drought tolerance in transgenic Arabidopsis lines varying in *ZAT18* expression [53].

## 3. Discussion

In this study, we used GWAS to dissect the genetic basis of GY, yield components, and GPC, under two water regimes in field conditions. The phenotypic results identified high variability among accessions for all traits, and statistical analyses indicated that 14 traits were significantly affected by water deficit. The thermal imaging of the field and the calculated stress index confirmed that plants in the D treatment were more stressed than in W. Most importantly, the variability between plots indicated that the collection included accessions with a variety of responses to drought. It is known that there are a general decrease under water stress conditions in GY and most yield components [57,58,59]. We found a reduction of GY and most yield components (SW, GWpS, GNpS, NSpP, TGW, GFE) which mostly exhibited high positive correlation among them, and negative correlations between GY and PH, DH, and GPC. A negative correlation between PH and GY could be attributed to fewer grains/ears in the tallest plants [60,61]. The different ratios of yield components to GY can be explained by the influence of the environment on plant growth [14]. This study shows that early maturation, small plant size, and reduced leaf area, are all expected to decrease plant photosynthetic activity, and were associated with drought tolerance. We found a reduction of GPC under D, a decrease in the quality and quantity of grains under water stress is associated with the impaired development of the reproductive organ [62,63]. Increasing the number of grains contributes to wheat yield potential under a limited water supply [64] and [14]. Moisture deficiency during flowering and at the time of ripening leads to impaired translocation of photosynthates to the grain and poor assimilation, forming respiratory losses [65]. 

The wheat panel comprised of landrace and modern cultivars of hexaploid wheat accessions from Europe, Asia, the Mediterranean, Africa, America, and Oceania, representing the worldwide eco-geographical range of grown cereals. The accessions were selected to identify trends in the frequency and location of sequence-predicted functional variants over geographical and environmental space [66]. Most of the accessions are cultivars that were bred for specific environments. We found that DH and GY were the main traits that could differentiate accessions based on their origin. GWAS analysis identified that 9 of 20 MTAs for DH reside in QTL clusters, or in relative proximity to MTA for a GY-component, and one QTL that included MTAs for GY and DH. Accessions from arid and Mediterranean climates, regions characterized by low precipitation and hot temperatures at grain filling, exhibited earlier DH and higher GY. Early reproduction (terminal drought escape) has been a successful breeding strategy for Mediterranean environments, which is possibly the reason for the association between early flowering and high GY in our experiment [67]. No less important are the 11 of 20 MTAs for DH that are not associated with GY or yield components, which are possibly associated with a drought resistance mechanism other than earliness. Combining earliness with additional drought resistance mechanisms may lead to greater drought resilience as compared to a single resistance mechanism. In contrast, accessions from temperate regions which are characterized by lower temperatures and high-water availability showed lower GY under both conditions. For example, the African accessions which had the shortest DH were the most stable between the two contrasting environments in terms of GY, while the European accessions required more DH and showed the greatest reduction of GY by 35%. Global climate change requires understanding the genetic basis of crop adaptability in response to drought [68]. Plant plasticity in response to drought is linked to changes in flowering phenology which impact GY [69,70,71]. The genomic architecture of the plasticity of wheat agronomic and physiological traits in response to drought demonstrated the effect of heading time on adaptation to varying water conditions [71]. The late flowering and different combinations of photoperiod sensitivity alleles in *Ppd*-A1 and *Ppd*-B resulted in reduced grain weight and GY [72]. PCA analysis of yield-related traits measured in the wheat panel enabled to identify of drought-adapted wheat accessions that outperformed under D or W conditions or in the two water regimes: (a) 15 accessions performed best under D conditions can be regarded as drought-resistant; (b) 9 accessions performed well under both drought and W conditions and can be regarded as drought resistant and stable between environments. Many of these 24 accessions originated from the Mediterranean (12) and Africa (5), but some high-yielding accessions under drought were from Asia (Japan, China, and India) and two from America (Mexico and Argentina). (c) the third group of sixteen accessions showed high GY under W can also be regarded as resilient accessions since they performed well in conditions of 450 mm. Annual precipitation of 450 mm is not regarded as high rainfall at their origin in Europe or America (e.g., Germany average of 780 mm; Bulgaria has an average of 670 mm annual rainfall; Switzerland has annual rainfall from 800 mm to 2400 mm). It was shown that in Portugal, such conditions at the most susceptible growth stage of post-anthesis and grain filling period can negatively affect GY [73]. The location of our experiment in the Israeli Northern Negev is warmer and drier than the typical Mediterranean climate and causes high evaporative demand in the late spring (ca. April-May) when precipitation is low. Thus, our analysis demonstrates the importance of evaluating the wheat panel in the hot and dry Israeli Northern Negev. The identified 40 accessions can be excellent candidates for cultivation in regions undergoing desertification.

GWAS identified 154 highly significant MTAs for nine of 14 traits (GY, DH, GWpS, GNpS, PedL, NSpP, PH, SW, and GPC). MTAs were found on most of the wheat chromosomes, but they were not equally distributed among the A-, B-, and D-genomes or within the chromosomes in each genome. The low number of SNPs mapped on genome D probably led to the identification of a very low number of MTAs on this genome. The low nucleotide diversity within the D-genome was described earlier [66,74,75]. We found that under D conditions, the distribution of MTAs between the A- and B-genomes was asymmetric—there were more MTAs on the B-genome compared to the A-genome (48 under D vs. 27 under W, *p* < 0.05). In allotetraploid plants such as wheat, genome asymmetry implies that the homoeologous genes on the A-, B-, and D-genomes each make differential contributions to various traits. Genome asymmetry was previously found in the proportion of domestication-related QTLs in the B- and A-genomes [76,77]. For example, the B-genome included many genes regulating ecological adaptation and tolerance to abiotic stress, e.g., genes involved in wax production that would affect drought tolerance, boron tolerance, tolerance to iron deficiency, low cadmium uptake, and resistance to herbicides. Genes associated with plant and spike morphology and other traits of the ‘domestication syndrome’ are more abundant in the A-genome [74]. The number of “domestication QTLs” mapped to the A-genome was twice the number on the B-genome, supporting the concept of “genome asymmetry” in the domestication of wheat [78].

In the current study, we identified that Chr. 6B is an important source for 20 MTAs, of which 13 MTAs clustered into nine QTLs were found under D. These 13 MTAs contributed 27% of all 48 MTAs found on the B-genome under D. Seven MTAs for yield were found in four adjacent QTLs (6B.10–6B.13) under D and/or W conditions. Under D, we found QTLs for GY at 6B.10 and SW at 6B.12. Under D, we also found QTLs for SW and the NSpP at 6B.12; QTLs for GY and DH at 6B.11; and DH at 6B.13. We suggest that this region can be regarded as a hotspot of genes influencing yield in response to low water availability. On Chr. 6B, we also found two MTAs for GPC, the first (6B.1) is novel, while the second MTA found at 6B.7 mapped to the same location as the previously cloned Gpc-B1, which is derived from wild emmer wheat [51]. This QTL was also found in two other mapping populations derived from wild emmer wheat [79]. The importance of Chr. 6B as a source of alleles for breeding was demonstrated by the locations of yield QTLs [80,81,82,83]. The 29 QTLs were identified in 179 recombinant inbred lines population tested in saline fields and determined that more than half of the loci were on chromosomes 2B and 6B, which are aligned with QTLs controlling the number of grains per ear and grain weight per ear [84]. The effect of heat stress on winter soft wheat, at flowering and grain filling, activated genes in the QTLs on Chr. 6B, which were associated with a response to post-anthesis heat stress and with the maintenance of thousand-grain weight under heat [85]. Interestingly, a meta-QTL analysis performed on 230 published reports from 1999–2020 for tetraploid wheat showed that the Chr. 6B accumulates QTL effects/genes for the main features that underscore GY [86]. In addition, the high-temperature yellow rust resistance gene *Yr36* was also cloned from Chr. 6B [87,88]. Sequencing of Chr. 6B revealed the location of the *Nor-B2* and *Gli-B2* genes that are important for homologous recombination and GPC, respectively [89]. Rice *OsGW2* is associated with grain width [90], and its orthologue in wheat, *TaGW2* cloned from Chr. 6A had SNPs in its promoter region associated with seed width and TKW [91]. Further haplotype association analysis in wheat showed that heading time and maturity date varied among modern cultivars between Hap-6B-1 and Hap-6B-4. *TaGW2* on Chr. 6B has a stronger effect on TKW than *TaGW2* on Chr. 6A, and Hap-6B-1 was the preferred grain width and weight-increasing haplotype that had undergone strong positive selection [82]. The strongest genetic diversity selection on *TaGW2*-6B occurred at the tetraploid level and was found in the promoter regions [92]. Several studies in wheat were conducted to confirm the association of *TaGW2* homologous in wheat and to confirm its function as it was known in rice as a negative regulator for grain width [93,94]. Overall, the down-regulation of *TaGW2* copies resulted in smaller wheat kernels, thus suggesting that *TaGW2* is a positive regulator of grain size-related traits [90]. Generated mutations in the homoeologous copies of the *TaGW2* by gene editing and TILLING, confirmed its negative regulation mechanism and showed that in *TaGW2*-6B mutant caused the highest single-genome increases in grain size in the cultivar [95]. Searching the physical location of *TaGW2*-6B in our GWAS study shows that it is found downstream of QTL 6B.6 MTA for SW under W (physical location 252094108) as compared with its location in the Norin cultivar. Many candidate genes were found for MTAs on Chr. 6B were associated with stress response are described in Appendix A, the above information supports our GWAS results suggesting that Chr. 6B is an important source for yield-related genes underlines important QTL for breeding. 

## 4. Materials and Methods

### 4.1. Plant Materials

The current study consisted of 232 bread wheat (*Triticum aestivum*) accessions selected from a larger global wheat diversity panel comprising 487 accessions of hexaploid and tetraploid wheat, that was assembled by the European consortium (Whealbi) and described previously by [66]. The passport data of the full wheat panel can be retrieved through the URGI portal (https://wheat-urgi.versailles.inra.fr/Projects/Whealbi (accessed on 1 January 2019)). We first confirmed that plants have spring-type phenology under the Israeli winter conditions by growing them in 2015–2016. For the current 2016–2017 experiment we used spring-type bread wheat accessions, including landraces, and modern cultivars from 54 countries of the five continents: Europe (10), Asia (93) (37 from the Mediterranean and 56 from East Asia countries), America (18), Africa (12) and Oceania (6). The passport data of 232 accessions and varieties used in the current study are presented in Appendix A. 

### 4.2. Field Experiment

Plants were grown in winter 2017 (December 2016 to May 2017) in Israel. The experiment was conducted in a homogenous field, at Urim farm at the Northern Negev region (31.324257; 34.532265), under two treatments: water-limited (D) and well-watered (W). The set up included two main plots, one per treatment, each consisting of three replicates in a randomized block design. Each experimental plot (1.2 m × 2 m) consisted of six 2 m long and 0.2 m spaced rows, sown at a rate of 170 seeds per m^2^. Precipitation during the growing season was 140 mm, which was supplemented for both treatments by sprinkles up to 250 mm, the amount designated for the D treatment. To avoid a spill of water from the W to the D treatment, drip irrigation was used to supplement the W treatment with additional water up to its designated amount, 450 mm (Appendix A). A 2 m wide border was sown between the main plots of the two treatments, thus avoiding penetration of groundwater or plant roots between plots. The average maximum temperature during grain filling varied from 23 °C (on 15th April to 15th May) and 30 °C (on 16th May to 15th June) (Appendix A). Weed, diseases, and insects were treated as needed, according to the recommendation for commercial wheat cultivation in that region. Additional biotic or abiotic stress which could affect the results cannot be controlled in such a large field experiment, however, if it occurs influences all accessions tested in the field. 

### 4.3. Phenotyping

The field plots were inspected twice weekly to determine days-to-heading (DH). A plot was determined as heading when the spike emerged for ¼ of its length in 50% of the plants (Zadoks DGS 53). Flag leaf length (LL, cm), Flag leaf width (LW, cm), and Flag leaf area of two plants/plot (LA = length × width × 0.75, were 0.75 is an empirical coefficient) and specific leaf weight (SLW = weight/LA) were assessed. Upon maturity, two plants of each plot were measured in the field for (a) plant height (PH, from the soil surface to spike top), (b) peduncle (upper internode) length (PedL; cm), and (c) spike length (SL; cm). Subsequently, 15 spikes were manually harvested from each plot and used to determine the average spike weight (SW; g). Spikes were threshed and grain weight per spike was calculated (GWpS; g), grains were counted to determine grain number per spike (GNpS) and thousand-grain weight (TGW; g), grain filling efficiency (GFE = GWpS/SW), and the number of spikes per plot (NSpP = grain yield/GWpS) were calculated. Osmotic potential (OP, MPa) was determined to sample leaves for 1–2 weeks after heading. Finally, the entire plot was harvested mechanically (Wintersteiger, Delta Plot combine) and grain was weighed to determine total grain yield (GY; g/m^2^). Grains extracted from the 15 spikes were also used to determine grain protein content (GPC, %). 

For analysis GPC, 1.5 gr. of seeds were ground using a Laboratory Mill 3310 (Perten a PerkinElmer company, Waltham, MA, USA). Then, the flour was tested for GPC by Perten Inframatic 9520 NIR Flour Analyzer (Perten a PerkinElmer company, Waltham, MA, USA). 

### 4.4. Thermal Imaging of Plant Canopy

An aerial thermal imaging campaign was conducted at midday on April 20th, 2017. A FLIR SC655 camera (FLIR^®^ Systems, Inc., Bilerica, MA, USA) was mounted on a 6-engine drone (Datamap Group, Bnei Brak, Israel). More details on the thermal camera, flight height and pre-processing can be found in [96]. During the imaging, a meteorological station was mounted in the field to measure the meteorological conditions. Image analysis and mapping were conducted using ArcGIS (ESRI, Ltd., Tokyo, Japan). Following the separation of canopy pixels from soil pixels [97] crop water stress index (CWSI) was calculated [96,97,98] as follows: CWSI = (T_canopy_ − T_wet_)/(T_dry_ − T_wet_). Where T_canopy_ is the temperature of the canopy in the thermal image, and T_dry_ and T_wet_ are two reference temperatures. T_dry_ was set as air temperature + 50 °C [99,100] and T_wet_ was calculated using the energy balance equation [101]. More details on the approach for calculating CWSI can be found in [97].

### 4.5. Statistical Analysis of Phenotype Data

All phenotypic data were normalized using the Shapiro–Wilk test before analysis. The problem of asymmetric distributions was addressed using the Box–Cox transformation in the R package (https://cran.r-project.org/web/packages/caret (accessed on 9 August 2022)). The statistical package (STATISTICA.V10, StatSoft. Inc. 2011, Tulsa, OK, USA) was used for descriptive statistics, and correlation for all statistical analyses unless specified otherwise. Sample homogeneity was determined by the deviation of the coefficient of variation (CV). Variables with a scatter threshold CV greater than 35% were excluded from the analysis. Analysis of variance (ANOVA) was used to assess the possible effects of accessions and environmental conditions. The results were considered significant at *p* < 0.05 or *p* < 0.01 and the mean comparisons were performed using Tukey’s honest significant difference (HSD) test. The variation of each trait under the two watering regimes was presented by employing the “ggplot2” package in R (https://cran.r-project.org/web/packages/ggplot2/index.html (accessed on 3 May 2022)). The squared cosine values were calculated from principal component analysis (PCA). Squared cosine values of traits and genotypes provide obvious estimates about them. If genotypes or traits having higher values (factor scores >70) in 1PC and lower values in 2PC are regarded as promising genotypes or traits irrespective of conditions. If a genotype has, a lower square cosine value in the 1PC and has a higher squared cosine value in 2PC regarded as a notorious genotype [62,102,103]. Correlation coefficients were calculated by Pearson’s method at a significance level of *p* < 0.05 using the “Corrplot” package (https://github.com/kwstat/corrgram (accessed on 29 April 2021)). BLUP values were calculated using Henderson’s matrix notation as follows: Y = Xβ + Zu + e. Where Y is the vector of observed phenotypes; β is an unknown vector containing fixed effects, including the genetic marker, population structure (Q), and the intercept; u is an unknown vector of random additive genetic effects from multiple background QTL for individuals/lines; X and Z are the known design matrices, and e is the unobserved vector of residuals. The u and e vectors are assumed to be normally distributed with a null mean and variances. Broad sense heritability (*H*^2^) in the individual environment was estimated by *H*^2^ = σ^2^_a/_σ^2^_a_ + σ^2^_e_. Where: *H*^2^—heritability; σ^2^_a_ as the additive genetic variance, and σ^2^_e_ is the residual variance in the “Heritability” package (https://cran.r-project.org/web/packages/heritability (accessed on 13 December 2019)) in R language package (www.r-project.org, (accessed on 10 March 2022)).

### 4.6. Genotyping

As described above, the global hexaploid and tetraploid wheat panel was assembled by the European consortium (Whealbi). The genotyping of this collection was conducted within the framework of the consortium. All the information on DNA extraction, exome design, exome sequencing, and variant calling are available in [66]. As members of Whealbi consortium we used genotyping of 232 accessions for GWAS. Briefly, SNPs were generated by Exome capture (NimbleGen SeqCap EZ Exome kit), and sequencing was performed using the Illumina HiSeq2500 high throughput model, providing an average of 34 million read pairs per genotype. The genotyping data obtained for the full Whealbi exome capture sequencing result included ~390,000 SNPs. SNP data that were excluded from GWAS analysis included: minor allele frequency (MAF) of less than 5%, and SNPs that had more than 25% missing calls.

### 4.7. Population Structure and Kinship Matrix

The population structure of the wheat panel was assessed using the Bayesian clustering method in STRUCTURE version 2.3.4 [104]. Population structure was determined by inferring K from two to ten using 5000 burn-in iterations followed by 10, 000 MCMC (Markov-Chain Monte Carlo) iterations and five replications for each K. For analysis, clusters Evanno’s correction method was applied [105]. The obtained results were analyzed using STRUCTURE harvester (http://taylor0.biology.ucla.edu/structureHarvester (accessed on 1 July 2014)) to get the appropriate K value. Information about relationships is conveyed through the kinship (K) matrix, which is used in a mixed linear model (MLM) as the variance-covariance matrix between the individuals. To improve the statistics and power of the analysis, a kinship matrix (K) based on genetic markers was used in conjunction with the population structure (Q+K).

### 4.8. Genome-Wide Association (GWAS) and Identification of Marker-Trait Associations (MTAs)

For GWAS we used the R package Genome Association and Prediction Integrated Tool (GAPIT) (http://www.maizegenetics.net/GAPIT (accessed on 4 May 2022)) [23]. Five GWAS models were used: General Linear Model (GLM), MLM, Mixed Linear Multiple Loci Model (MLMM), Fixed and Random Circulating Probability Unification Model (FarmCPU), and Bayesian information and Linkage disequilibrium Iteratively Nested Keyway (BLINK). The population structure analysis included a PCA matrix (P), and a kinship matrix (K) were tested in GAPIT. The significance of associations between markers and phenotypes (MTA) was assessed using the false discovery rate (FDR) [106]. The best model for each trait was chosen based on the distribution of the *p*-values using QQ plots [107]. The chosen best model was through the maximum of the observed *p*-values with the expected ones with a minimum deviation from the trend made it possible to choose models that more fully describe the MTA [107]. MTAs were identified at an FDR-adjusted *p*-value of 0.05, except for MTAs where the threshold was set to 0.1 due to low and moderate *H^2^*. A region of ~1 Mbp that contains a few adjacent MTAs (clusters of MTAs) was regarded as a QTL region. Manhattan plots and SNP-density plots were generated by rMVP package [108]. Gene annotation of *T. aestivum* cv. Chinese Spring (IWGSC RefSeq annotation v2.1) was used to retrieve high-confidence genes surrounding each MTA, in a window of ±250 Kb.

## 5. Conclusions

The present GWAS identified significant genetic polymorphism and high phenotypic diversity in the response to water stress. Most of the measured phenotypic traits were negatively affected by water stress. Our analysis identified accessions, which mainly originate from the Mediterranean and Africa, that show high GY under water-limited conditions or with stable yield in W and D conditions. These excellent wheat accessions can be recommended for cultivation in regions experiencing low precipitation and increased temperatures. GWAS identified 154 MTAs, partly residing in clusters in QTL regions, that have a significant effect on GY, or yield components, and GPC under water-limited and/or well-watered conditions; some of them can be used for breeding. A higher number of MTAs identified under D were found in genome B than in A or D genomes, with Chr. 6B as the main contributor. Based on the physical location of the MTAs, we identified promising candidate genes underlying yield-related traits. 

## Figures and Tables

**Figure 1 ijms-23-10575-f001:**
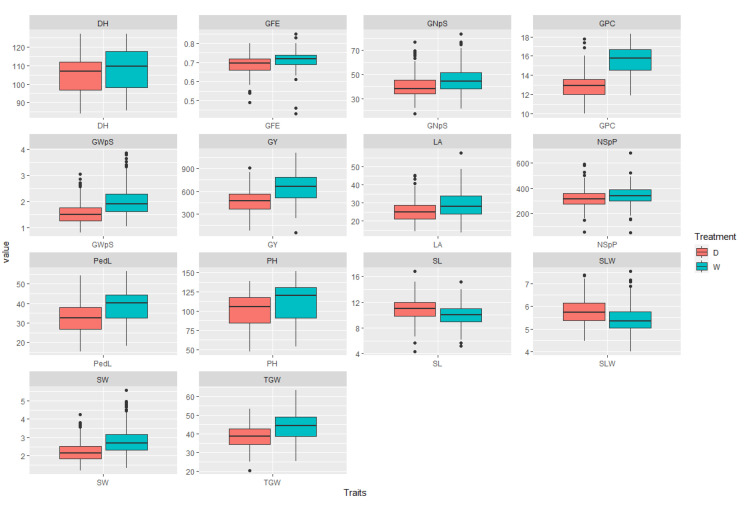
Comparison of averages of phenotypic traits in the wheat panel of 232 accessions, measured under well-watered (W) and water-limited (D) conditions. Key of traits and scales measured units: grain yield (GY, g/m^2^), grain protein content (GPC, %), spike weight (SW, g), grain weight per spike (GWpS, g), grain number per spike (GNpS, #), number of spikes per plot (NSpP, #), thousand-grain weight (TGW, g), the efficiency of grain filling (GFE, #), leaf area (LA, cm^2^), specific leaf weight (SLW, mg/cm^2^), plant height including spike (PH, cm), peduncle length (PedL, cm), spike length (SL, cm), days-to-heading (DH, #).

**Figure 2 ijms-23-10575-f002:**
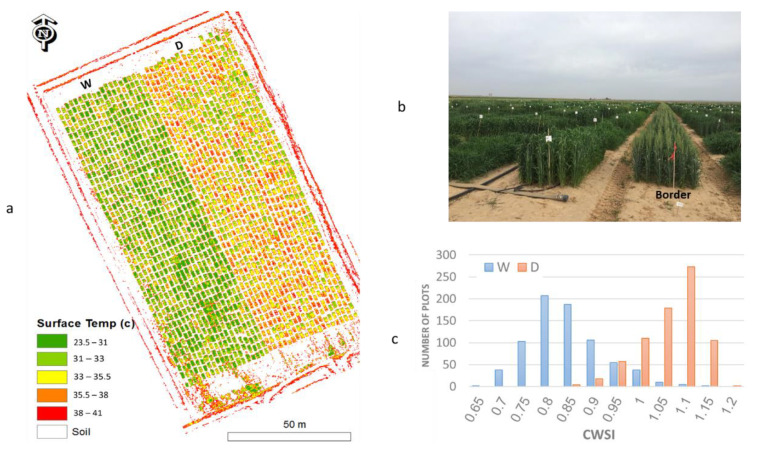
(**a**) Surface temperature map of the experimental field consisting of well–watered (W), and water-limited (D) treatments; (**b**) Experimental field, where the border between two water regimes is marked with red flag; (**c**) Distribution of crop water stress index (CWSI) within the two treatments.

**Figure 3 ijms-23-10575-f003:**
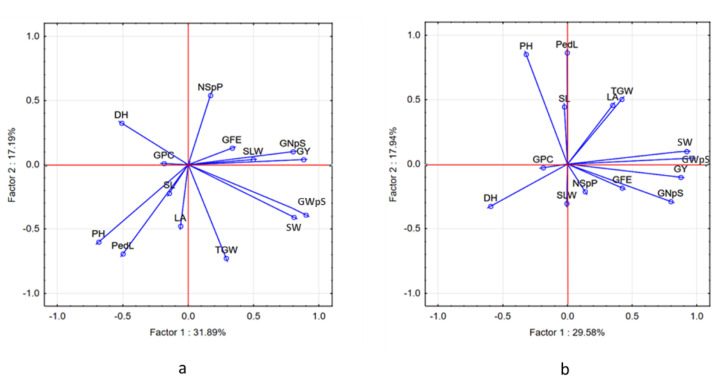
Principal component analysis (PCA), and biplot vectors of 14 traits measured in the 232 wheat accessions in two water regimes: (**a**) well-watered (W) and (**b**) water-limited (D). Key: grain yield (GY), grain protein content (GPC), spike weight (SW), grain weight per spike (GWpS), grain number per spike (GNpS), number of spikes per plot (NSpP), thousand-grain weight (TGW), the efficiency of grain filling (GFE), leaf area (LA), specific leaf weight (SLW), plant height including spike (PH), peduncle length (PedL), spike length (SL), days to heading (DH).

**Figure 4 ijms-23-10575-f004:**
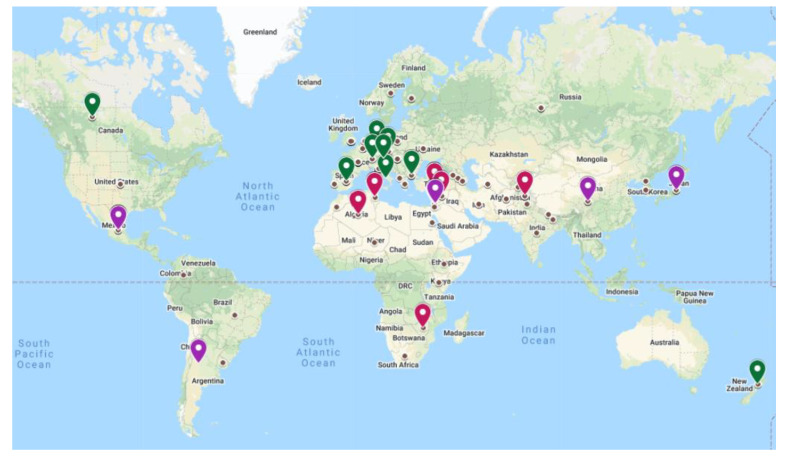
Geographic origins of the best performing accessions. Accessions with high GY in D (red); in D + W (purple); in W (green). The origins of the 232 accessions are marked as small brown circles.

**Figure 5 ijms-23-10575-f005:**
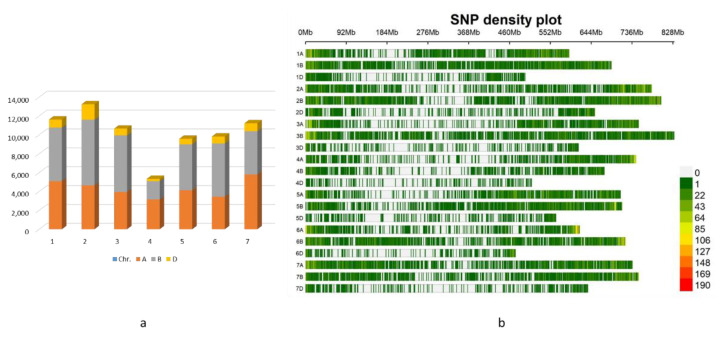
Distribution and density of SNPs mapped to the wheat genome: (**a**) Distribution of SNPs on 7 chromosomal groups, and the number of mapped SNPs on each wheat genome A, B, and D; (**b**) Density plot of SNPs mapped to the different wheat chromosomes.

**Figure 6 ijms-23-10575-f006:**
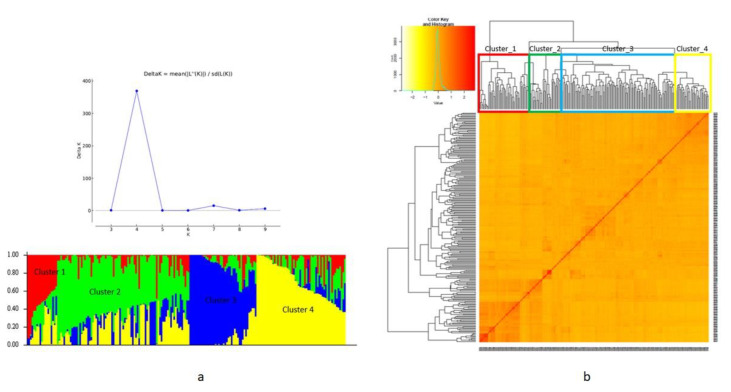
Population structure and kinship matrix of the wheat diversity panel based on 71,571 SNP markers: (**a**) STRUCTURE analysis and plot of delta K (1 to 10), and the presence of a peak at K = 4 hint at four subgroups. The 1st cluster (red), 2nd cluster (green), 3rd cluster (blue), and 4th cluster (yellow); (**b**) Heat map of the identity-by-descent based on genomic relationship matrices (GRM).

**Figure 7 ijms-23-10575-f007:**
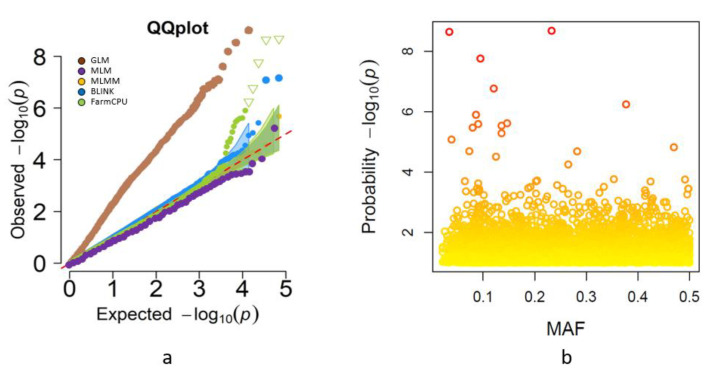
Multilocus model for comparing GY in the wheat panel: (**a**) Quantile-quantile (QQ) plot of five GWAS models (GLM, MLM, MLMM, BLINK, and FarmCPU) showing the expected versus observed -log10 (*p*-value) of each SNP marker (shown as dots). The red line is the expected distribution under the null hypothesis; (**b**) Minor allele frequency (MAF) plot –log_10_(*p*). The *p*-values for each SNP marker were plotted against their MAF for the FarmCPU model.

**Figure 8 ijms-23-10575-f008:**
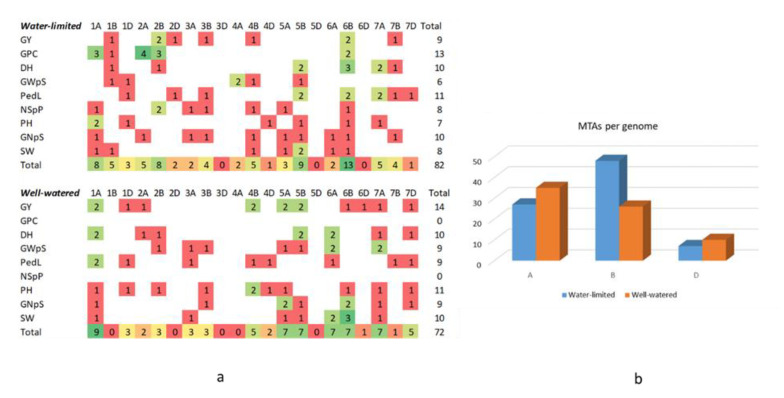
The distribution of MTAs for the different traits were revealed under water-limited (D) and well-watered (W) conditions: (**a**) MTAs in 21 wheat chromosomes; the colors represent an increasing number of MTAs for each trait (e.g., 1 MTA (red), 2 MTAs (green); 3 MTAs (dark green)); (**b**) distribution of MTA distribution under D and W conditions, in A, B, and D genomes of wheat.

**Figure 9 ijms-23-10575-f009:**
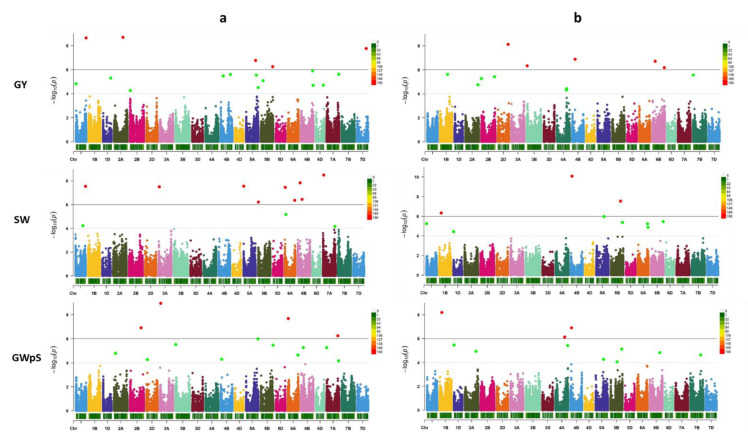
Manhattan Plots across all 21 wheat chromosomes, for nine traits: GY—Grain yield; SW—spike weight; GWpS—grain weight per spike; NSpP—number of spikes per plot; PH—plant height; PedL—peduncle length; GNpS—grain number per spike; DH—days to heading; GPC—grain protein content. SW, GNpS, GWpS, NSpP, PH, PedL, DH (FDR < 0.05), and GPC (FDR < 0.1). The (**a**) well-watered (W), and (**b**) water-limited (D) conditions. The FarmCPU model was used for GY, SW, GNpS, GWpS, NSpP, PH, PedL, DH, and the GLM model for GPC. The two horizontal lines indicate 10^−4^ and 10^−6^ thresholds of significance. The *x*-axis located 21 chromosomes in wheat. The *y*-axis located *p*-values (−log transformed).

**Table 1 ijms-23-10575-t001:** Descriptive statistics of phenotypic traits measured under varying water supply in 232 accessions. Mean, minimum (Min), and maximum (Max) values, standard deviation (SD), coefficient of variance (CV), and heritability estimates (*H*^2^) for grain yield (GY), grain protein content (GPC), spike weight (SW), grain weight per spike (GWpS), grain number per spike (GNpS), number of spikes per plot (NSpP), thousand-grain weight (TGW), the efficiency of grain filling (GFE), leaf area (LA), specific leaf weight (SLW), plant height including spike (PH), peduncle length (PedL), spike length (SL), days-to-heading (DH), Osmotic potential (OP), Flag leaf length (LL), Flag leaf width (LW), (# = number).

Water-Limited (D)							Well-Watered (W)					
Traits	Mean	Min	Max	SD	CV	*H* ^2^	Mean	Min	Max	SD	CV	*H* ^2^
GY, g/m^2^	480.85	83.64	908.04	160.90	33.46	0.75	659.49	55.21	1107.81	206.14	31.26	0.71
GPC, %	12.95	10.00	17.80	1.26	9.73	0.20	15.61	11.90	18.30	1.39	8.95	0.10
SW, g	2.27	1.22	4.38	0.60	26.48	0.55	2.86	1.35	5.56	0.78	27.17	0.89
GWpS, g	1.56	0.63	3.04	0.44	28.31	0.55	2.02	1.03	3.86	0.58	28.54	0.76
GNpS, #	40.13	17.33	76.67	9.74	24.28	0.36	45.82	21.67	83.67	11.04	24.10	0.41
NSpP, #	314.19	55.00	586.33	77.02	24.51	0.56	334.99	50.33	678.33	76.93	22.96	0.45
TGW, g	39.03	20.33	57.87	6.46	16.54	0.12	44.50	25.27	71.13	7.96	17.88	0.34
GFE, #	0.69	0.37	0.80	0.06	8.12	0.37	0.71	0.43	0.85	0.05	7.73	0.39
LA, cm^2^	25.91	9.42	45.10	6.34	24.46	0.46	29.58	13.50	57.71	7.51	25.39	0.28
SLW, mg/cm^2^	5.78	4.45	7.40	0.58	10.13	0.21	5.39	4.00	7.54	0.62	11.56	0.21
PH, cm	101.15	47.50	139.00	21.12	20.88	0.75	112.84	53.83	152.67	23.75	21.05	0.87
PedL, cm	32.84	15.33	54.33	7.75	23.61	0.67	38.96	18.17	56.83	8.91	22.87	0.70
SL, cm	10.88	4.33	16.75	1.75	16.09	0.25	9.86	5.17	15.17	1.61	16.31	0.35
DH, #	106.09	84.00	127.00	10.88	10.25	0.46	108.07	85.67	127.00	10.37	9.59	0.99
OP, MPa	−2.12	−2.81	−1.55	0.23	11.11	0.16	−1.64	−2.15	−1.27	0.20	11.09	0.17
LL, cm	20.62	11.87	30.58	3.49	17.00	0.32	22.85	15.05	35.45	4.03	17.62	0.29
LW, cm	1.65	0.87	2.23	0.21	12.69	0.20	1.70	1.16	2.28	0.23	13.61	0.13

## Data Availability

All data supporting the reported results are included within the article or its Appendix A.

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
