# Peer review of "Genome-Wide Association Study in Bread Wheat Identifies Genomic Regions Associated with Grain Yield and Quality under Contrasting Water Availability"

_ijms, 2022, doi:10.3390/ijms231810575_

Round 1

Reviewer 1 Report

General comments

This paper is a very important study for understand thing future possibilities of agriculture under climate change. It has important implications and recommendations for which accessions to grow in water restricted areas in the future. Methods require some clarification, especially regarding BLUPs, which I think is expected from GWAS papers. I recommend this paper for publication after some revisions.

Introduction

Very well written. Good summary of traits.

Line 94. Can you reference a general or fundamental paper for CRISPR?

Lines 101-114 are a summary of results rather than what I would expect should be a summary of the aims of the study.

Methods

Table S1, what is the meaning of “BAD” genome?

Line 592. Approximately how many plants per row or plot?

Can you be confident that treatments / water did not flow between D and W blocks? (I do not work in broadacre crops, so please excuse my ignorance).

LA and SLW were calculated on how many leaves per plant/row/plot?

Line 608. Please provide more details on how GPC was determined from NIR.

Why were correlations conducted using STATISTICA and R?

Line 636. Please define the linear mixed model used to obtain BLUP values using common vernacular y = XB + Zg + e.

Line 638. Please show the calculation used by the package for estimating H2.

Line 646. Please clarify the wording for filtering SNPs, e.g. SNPs with less than 5% MAF and more than 25% missing calls were excluded.

Line 651. I don’t think that 1,000 burn in for STRUCTURE is usually enough: usually 5,000 or 10,000. Please either re-run, or include a plot as a supplementary figure showing that 1,000 is enough and that the results have stopped oscillating at that point.

Line 661. Do you have any reasons for choosing those specific GAPIT models?

Line 666. You haven’t stated that MTA actually stands for marker-trait associations.

Line 667. I wouldn’t say the correlation between p-values should be used to find the “best” model. Rather, I believe it is more common to compare the results of different models and see what the common significant SNPs are. Consider rewording.

Please include methods and more details for comparing phenotypes with ecogeographic origins. Please outline your reasoning for doing this – is it based on the original location and thus evolutionary genetics? When looking at the continental scale, there is a very broad range of environments. For example, in Oceania, the temperate environment of New Zealand is very different to the Mediterranean environment found near Perth in Western Australia and the desert of inland Australia. The interpretation of your results should be focussed on your aim of this analysis.

Results

It is not clear to me which 14 of the 17 traits were significantly difference between D and W. Should this be in Table 1 or Figure 1? You have only shown the significant 14 traits in Table S2, but I suggest including all 17 even though 3 were not significant.

Figure S2 description needs clarifying. 3 correlations are shown in the upper diagonal – overall, D and W. Diagonal is the distribution of phenotypes, and the lower diagonal are scatter plots.

I would suggest including the H2 values in a table. H2 of 0.99 was for which trait? This seems unlikely for a quantitative trait. This result should be discussed and compared to other studies.

Figure S3 is quite confusing for me, especially the black lines. Please confirm “n” for not significant, as this does not appear correct to me based on the distribution of data points, especially for part e. Part d has black lines indicating significant differences, but also has “n”s for not significant, and is this quite confusing for me.

Line 232. Cosine squared – I could not find details in the methods for this, and I am unsure of your reasoning and background for this analysis.

Lines 243-245. What are the p-values for, what are you comparing here?

For the accessions with NA origin, can you allocate them to a likely region based on STRUCTURE results? Was this of interest to you?

Line 280. You suggest that accessions have a high degree of relatedness, but what is your evidence for this?

Line 289. Why did you use 3 PCAs in your model (again, another reason to outline your BLUP LMM structure)? Did you perform a Wald test for fixed effects to see if all 3 were significant? Might 1 or 2 have sufficed?

Figure 7a. Please change the colours for either FarmCPU or MLM, as they are too similar at present.

Line 309. False associations can be due to not accounting for population structure, or pleiotropy. Suggest looking at Yang et al. 2011 European Journal of Human Genetics 19(7): 807-812 and Korte and Farlow 2013 Plant Methods.

Figure 8 caption. Please clarify that distribution is amongst chromosome and genomes.

Discussion

Line 451. Variability of accessions within plots – please reword “plots” as can be confused with experimental design.

Lines 452-453. Should this read “and of most yield components”?

Line 458. “These studies” – which studies are you referring to?

Lines 472-474. You mention the relationship between GY and DH. On line 349, you say that most MTAs for GY did not map near those for DH, but they did for other component traits. Can you discuss the relationship between GY and DH with regard to the MTAs not being co-located?

Lines 487-491 is unclear to me. “3 types … or in the 2 water regimes”. Please clarify, e.g. what are the 3 types.

I think it is worth mentioning that under climate change, it is predicted that we will experience more extreme weather events. So whilst there may be overall more drought conditions, there may also be more heavy rainfall events and storms. Will growers and breeders need to be aware of this when growing your chosen accessions?

Author Response

File attached

Reviewer 2 Report

In line 87, the widely used GWAS statistical methods should be mentioned, and the corresponding references should be added. The following references should be added.

MLM-based methods

Yu, J. et al. A unified mixed-model method for association mapping that accounts for multiple levels of relatedness. Nat. Genet. 38, 203–208 (2006).

Kang, H.M. et al. Efficient control of population structure in model organism association mapping. Genetics 178, 1709–1723 (2008).

Zhou, X. & Stephens, M. Genome-wide efficient mixed-model analysis for association studies. Nat. Genet. 44, 821–824 (2012).

Zhang, Z. et al. Mixed linear model approach adapted for genome-wide association studies. Nat. Genet. 42, 355–360 (2010).

Kang HM, Sul JH, Service SK, et al.  Variance component model to account for sample structure in genome-wide association studies. Nat Genet 2010; 42:348–54.

Lippert, C., Listgarten, J., Liu, Y., Kadie, C. M., Davidson, R. I., & Heckerman, D. (2011). FaST linear mixed models for genome-wide association studies. Nature methods, 8(10), 833-835.

Multi-locus MLM methods

Segura, V. et al. An efficient multi-locus mixed-model approach for genome-wide association studies in structured populations. Nat. Genet. 44, 825–830 (2012).

Wang, S. B. et al. Improving power and accuracy of genome-wide association studies via a multi-locus mixed linear model methodology. Sci. Rep. 6, 19444 (2016).

Wen, Y. J. et al. Methodological implementation of mixed linear models in multi-locus genome-wide association studies. Brief. Bioinform. bbw145

In line 294, authors should explain how to determine the optimal model for each trait.

It is better to add the Manhattan Plots of the GLM, MLM, MLMM and BLINK in the supplements.

In figure 1, statistical tests should be added to determine whether there are statistically significant differences in the mean values of specific features between well-watered and water-limited. And the significances should be marked in the figure.

In Figure 7, The QQ plot is only for GY trait. It is better to add QQ plots for other traits in the supplements.

In figure 9, the axis titles of the x-axis and y-axis are not clear. Also, the labels of x-axis and y–axis are not clear. The size of them should be larger.

In figure 9, please explain why 10-4 and 10-9 were chosen. Do you think the threshold using Bonferroni correction is a better choice than a fixed threshold?

Author Response

File attached

Reviewer 3 Report

The study is based on a wide number of accessions but the methods employed have some limitations:

- As far as I understand from the text (it is not clear enough) accessions were grown in an experimental field. If so, other factors (besides drought) might have influenced the results. How was the microbiome, insects or other abiotic conditions controlled?

- The implementation of drought is not explained. Only the abstract reports that accessions were grown under "two environments: water-limited (D; 20 250 mm), and well-watered (W; 450 mm) conditions". This is crucial to follow the results.

- Details are missing about extraction and how sequencing was performed. Data quality analyses are also missing. Also, the reference genome mentioned in the results should be better explained in M&Ms to define how read mapping was performed.

- The number of plants used to quantify the traits should also be mentioned.

Author Response

File attached

Round 2

Reviewer 1 Report

All of my comments were addressed in a comprehensive manner. Thank you. 

Author Response

Dear reviewer,

Thank you very much for your excellent review which enabled us to improve our manuscripts!

Reviewer 3 Report

I appreciate the time and efforts of the authors to address the concerns that I raised in the previous version. Based on the new explanation, I feel that the experimental procedure can now be followed. However, it does have constraints- which were clearly highlighted by the authors in their answer - but not in the text itself. "Additional biotic (e.g. microbiome) or abiotic stress (high temperature) which could affect the results cannot be controlled in such a large field experiment, however, if it occurs it influences all accessions tested in the field. In our view, these uncontrolled factors contribute to the relevance of results obtained in field experiments, like the one in our study, for future application in breeding." 

This is indeed significant because the answers that the authors are reporting do not only reflect the influence of drought but also other factors (abiotic and biotic). This should be clearly stated, at least in the discussion.

In relation to the genomic data, the authors are now saying that " The Whealbi consortium was responsible for exome capture sequencing of this collection that is further used for GWAS of different traits, and also for data analysis.". Again, this should be clearly stated in the manuscript. The authors are placing much of these analyses and how they were performed based on a previous study, and it is not clear what information is new in this study. 

As a minor note, the pdf version that I received has multiple formatting errors and repetitive headings. I am sure that the authors can easily correct this.

Author Response

Dear reviewer,

As you suggested, we have added information to the description of field experiment and genotyping. We thank you very much for your review that helped us to improve the manuscript.